# Identification and Growth Characteristics of a Gluten-Degrading Bacterium from Wheat Grains for Gluten-Degrading Enzyme Production

**DOI:** 10.3390/microorganisms11122884

**Published:** 2023-11-29

**Authors:** Ga-Yang Lee, Min-Jeong Jung, Byoung-Mok Kim, Joon-Young Jun

**Affiliations:** Food Convergence Research Division, Korea Food Research Institute, Wanju-gun 55365, Republic of Korea; rkdid0925@gmail.com (G.-Y.L.); ooojmj@nate.com (M.-J.J.); bmkim@kfri.re.kr (B.-M.K.)

**Keywords:** wheat gluten, gluten intolerance, gluten-degrading bacteria, *Bacillus amyloliquefaciens* subsp. *plantarum*, α/β-gliadin protein

## Abstract

Immunogenic peptides from wheat gluten can be produced during digestion, which are difficult to digest by gastrointestinal proteases and negatively affect immune responses in humans. Gluten intolerance is a problem in countries where wheat is a staple food, and a gluten-free diet is commonly recommended for its treatment and prevention. Enzyme approaches for degradation of the peptides can be considered as a strategy for its prevention. Here, we isolated a gluten-degrading bacterium, *Bacillus amyloliquefaciens* subsp. *plantarum*, from wheat grains. The culture conditions for enzyme production or microbial use were considered based on gluten decomposition patterns. Additionally, the pH range for the activity of the crude enzyme was investigated. The bacterium production of gluten-degrading enzymes was temperature-dependent within 25 °C to 45 °C, and the production time decreased with increasing culture temperature. However, it was markedly decreased with increasing biofilm formation. The bacterium decomposed high-molecular-weight glutenin proteins first, followed by gliadin proteins, regardless of the culture temperature. Western blotting with an anti-gliadin antibody revealed that the bacterium decomposed immunogenic proteins related to α/β-gliadins. The crude enzyme was active in the pH ranges of 5 to 8, and enzyme production was increased by adding gliadin into the culture medium. In this study, the potential of the *B. amyloliquefaciens* subsp. *plantarum* for gluten-degrading enzyme production was demonstrated. If further studies for purification of the enzyme specific to the immunogenic peptides and its characteristics are conducted, it may contribute as a strategy for prevention of gluten intolerance.

## 1. Introduction

Wheat is one of the grains consumed as a staple food, with an annual consumption of approximately 66.09 kg per person worldwide [1]. Wheat grains contain 8–16% protein, out of which gluten accounts for 80–90% [2]. Gluten proteins play critical roles in food and baking industries because of their viscoelastic properties [3,4]. Some gluten proteins form immunogenic peptides during hydrolysis, which are poorly digested by the proteases of the gastrointestinal tract, such as pepsin, trypsin, and chymotrypsin, and can cause disease in sensitive people [5]; immunogenic peptides can cause disorders of the human immune system, such as celiac disease, bakers asthma, wheat allergy, and non-celiac gluten sensitivity [6]. The peptides generated during gluten hydrolysis are deaminated by tissue transglutaminase, which leads to the stimulation of the immune system [3]. The deaminated peptides may make a weak intestinal wall, the villi are blunt or broad, and these symptoms lead to malnutrition with reduction in the ability to absorb nutrients [7].

Gluten proteins can be divided into two fractions according to their solubility in alcohol: gliadins (alcohol-soluble fraction) and glutenins (alcohol-insoluble fraction). The alcohol-soluble fraction includes α-, β-, γ-, and ω-forms of gliadin proteins, while the alcohol-insoluble fraction includes low- and high-molecular-protein subunits on the polyacrylamide gel electrophoresis [3,8]. In particular, 33 mer peptides derived from α-gliadin and 26 mer peptides derived from γ-gliadin are strong T cell stimulators and induce negative immune reactions, which are associated with gluten intolerance [9,10]. Gluten intolerance is commonly treated and prevented by a gluten-free diet, and gluten-degrading enzymes, luminal gluten binders, vaccines, and transglutaminase inhibitors are considered as major therapies [11,12].

As the gliadin immunogenic peptides are resistant to digestion by human gastrointestinal enzymes, specific enzymes are required for their digestion [2]. Some studies have reported the presence of gluten-degrading enzymes in bacteria, fungi, plants, and insects; the enzymes completely hydrolyze gluten proteins, including T cell epitope peptides [13,14]. Among these, microbial enzymes have attracted attention in food and pharmaceutical industries, and the market share of microbial enzymes is rapidly growing owing to various advantages, such as short processing time, cost efficiency, and eco-friendliness, compared to mammalian enzymes [15]. The immunogenic peptides are formed during the hydrolysis of proline/glutamine-rich prolamines, which can be hydrolyzed by gluten-degrading serine proteases [2,3], such as prolyl oligopeptidases, subtilisin, and sedilisin, which are gluten-degrading enzymes produced by *Sphingomonas* sp., *Flavobacterium* sp., *Myxococcus* sp., *Aspergillus* sp., *Flammulina* sp., *Chryseobacterium* sp., *Bacillus* sp., and *Streptomyces* sp. [16,17,18].

Effective microorganisms are occasionally discovered in fermented foods. They can hydrolyze the composite materials in fermented foods in addition to producing secondary metabolites that exhibit health-beneficial effects. Some gluten-degrading bacteria, *Bacillus* sp., *Enterococcus mundtii,* and *Wickerhamomyces anomalus*, have been found in fermented sourdoughs [17,19]. In this study, we isolated a gluten-degrading bacterium from the wheat grains fermented using a gluten-containing medium, identified the species, and investigated growth characteristics of the bacterium and the pH range at which the crude enzyme is active. In addition, decomposition patterns of gluten proteins and the presence of immunogenic proteins related to α/β-gliadins were analyzed.

## 2. Materials and Methods

### 2.1. Isolation of Gluten-Degrading Bacterium

Dried wheat grains (*Triticum aestivum*) were obtained from Daehan Flour Mills Co., Ltd. (Seoul, Republic of Korea). The wheat grains were added to sterile deionized water for swelling for 1 h, allowed to drain naturally, and crushed using a roll mill. The pericarp was removed using a 120 µm sieve and the wheat powder was incubated in ten-fold tryptic soy broth (TSB; Difco; Becton Dickinson and Co., Sparks, MD, USA) at 30 °C for 12 h. Then, the suspension was spread on tryptic soy agar (TSA; Difco) containing 0.5% gluten and incubated at 35 °C for 24 h. A clear zone on the agar indicated the decomposition of gluten, and colonies forming a clear zone were isolated. A pure culture of the bacterium on a 0.5% gluten and 0.1% glucose agar plate was prepared to stock the bacterium.

### 2.2. Morphological Observation of Gluten-Degrading Bacteria Using a Scanning Electron Microscope

Morphology of the bacterium was observed using a scanning electron microscope (SEM; Nanoeye, SNE-3000M; SEC Co., Ltd., Suwon, Republic of Korea). An overnight culture (approximately 6 log CFU/mL, TSB, 30 °C, and 15 h) of the bacterium was transferred to fresh TSB and incubated at 30 °C for 8 h. Then, 100 µL of the bacterial culture was filtered using a sterile polycarbonate membrane (0.2 µm pore; Advantech Toyo Kaisha Ltd., Tokyo, Japan) and washed with phosphate-buffered saline (PBS; pH 7.2). The bacterium was fixed with 100 µL of 2% glutaraldehyde for 30 min, washed with PBS (pH 7.2), dehydrated in a graded ethanol series (25–95%), added with t-Butyl alcohol, allowed to stand for 30 min, and freeze-dried. After platinum coating, the bacterium was observed at ×10,000 magnification using SEM set at 30 kV.

### 2.3. Identification of Gluten-Degrading Bacterium

16S ribosomal DNA sequencing and identification of the bacterium were performed by the Korean Culture Center of Microorganisms (KCCM, Seoul, Republic of Korea). Briefly, chromosomal DNA was extracted from the bacterium using the Wizard Genomic DNA Purification Kit (Promega Co., Madison, WI, USA). The DNA library was amplified using the MyCycler thermal cycler (Bio-Rad Laboratories Inc., Hercules, CA, USA) and the following primers: forward, 27F 5′-AGAGTTTGATCATGGCTCAG-3′ and reverse, 1492R 5′-GGATACCTTGTTACGACTT-3′ [20]. The amplicon was purified using a Wizard SV gel and PCR clean-up system (Promega Co.) and analyzed using the ABI PRISM3730XL analyzer with the ABI PRISM BigDye^TM^ terminator cycle sequencing kit (Applied Biosystems Co., Waltham, MA, USA). The bacterial species was identified using BLASTn (https://blast.ncbi.nlm.nih.gov/Blast.cgi, accessed on 1 April 2020) available at the National Center for Biotechnology Information (NCBI). To construct a phylogenetic tree, nucleotide sequences were downloaded from NCBI, aligned, and compared with nucleotide sequences retrieved from GenBank using ClustalW. The phylogenetic tree was generated using the neighbor-joining method [21] in MEGA software (version 7.0; Biodesign Institute, Tempe, AZ, USA) [22]. The percentage of replicate trees in which the associated taxa clustered together in the bootstrap test (1000 replicates) is shown next to the branches.

### 2.4. Examination of the Culture Condition for Gluten-Degrading Enzyme Production

The temperature and time of the bacterial culture were assessed to consider a suitable culture condition for the production of gluten-degrading enzymes. An overnight culture of the bacterium (TSB, 30 °C, 12 h) was inoculated into fresh TSB (approximately 6 log CFU/mL) and incubated for 48 h at 25, 35, and 45 °C. During incubation, the cultures were collected at regular intervals (0, 3, 6, 9, 12, 18, 24, 36, and 48 h) and analyzed for bacterial cell count, pH, and gluten-degrading enzyme activity.

The bacterial cell count was determined with the plate count method using TSA [23]. To measure the pH, the culture was centrifuged at 10,000× *g* for 5 min, and the resulting cell-free culture was used to measure the pH using a pH meter (SevenEasy S20K; Mettler Toledo International Inc., Columbus, OH, USA). Gluten-degrading enzyme activity was evaluated using the agar well diffusion method [24] with slight modifications. The cell-free culture was concentrated in a centrifugal evaporator tube (Amicon^®^ Ultra-15 centrifugal filter unit; Merck, Darmstadt, Germany). Then, 50 µL of the aliquot was added to each well (5 mm diameter) of the plate containing 0.5% gluten, and 1.5% agar and incubated at 30 °C for 12 h. The diameter of the clear zone (DCZ), which indicates enzyme activity, was measured using a Vernier caliper.

### 2.5. Biofilm Formation of Gluten-Degrading Bacterium

The biofilm formation of the bacterium was measured with the microtiter dish biofilm formation assay of O’Toole [25] with slight modification. In brief, the overnight culture (35 °C, 12 h) was inoculated into TSB to the final cell density of approximately 6 log CFU/mL. A total of 100 µL of the broth was transferred into a flat-bottom 96-well microplate (non-coated polystyrene) and incubated for 48 h at 25, 35, and 45 °C (collection intervals: 0, 3, 6, 9, 12, 18, 24, 36, 48 h). After incubation, the culture medium containing planktonic bacterial cells was discarded. The community of the attached cells to the well surface was suspended with 200 µL of 50% dimethyl sulfoxide and measured at 595 nm.

### 2.6. Analysis of Bacterial Carbohydrate Fermentation and Enzyme Activity

Bacterial carbohydrate fermentation and enzyme activity were screened using the API 50 CHB/E medium kit (BioMérieux, Marcy-l’Étoile, France) and API ZYM kit (BioMérieux), respectively, according to the manufacturer’s instructions. Briefly, bacterium was pre-cultured in TSB at 30 °C for 12 h, the kits were used, resulting samples were incubated at 30 °C for 24–48 h, and the results were interpreted.

### 2.7. Analysis of Decomposition Patterns of Gluten Proteins

The decomposition patterns of gluten proteins were analyzed using sodium dodecyl sulfate–polyacrylamide gel electrophoresis (SDS-PAGE) [18] with slight modifications. An overnight culture of the bacterium (TSB, 30 °C, 12 h) was inoculated into the fresh TSB (approximately 6 log CFU/mL) supplemented with either 1% gluten or 1% gliadin and incubated at 25, 35, and 45 °C for 0, 12, 24, and 48 h. The protein concentration in the culture was determined using the BCA protein kit (Biovision Inc., Milpitas, CA, USA). The culture was mixed with a buffer (pH 6.8; 62.5 mM of Tris-HCl, 2.5% glycerol, 2% SDS, 0.01% bromophenol blue, and 5% β-mercaptoethanol) at a ratio of 1:1 (*v*/*v*). Then, aliquots (10 µL) were loaded on a 12% Mini-PROTEAN^®^ TGX^TM^ Precast Gel (#4561045; Bio-Rad Laboratories Inc.) in Tris/glycine/SDS buffer (pH 8.3; #1610732; Bio-Rad Laboratories Inc.) using the Mini-PROTEAN^®^ Tetra System (Bio-Rad Laboratories Inc.). Gluten and gliadin (3 mg/mL) were used as standards, and a protein marker ranging from 10 to 170 kDa (EBM-2000; ELIPIS Biotech Inc., Daejeon, Republic of Korea) was used. Electrophoresis was performed at 150 V for 30 min. Subsequently, the loaded samples were stained with a Coomassie brilliant blue solution (30% methanol, 10% acetic acid, and 1% Coomassie brilliant blue G-250). After decolorization, the gel was visualized using the Gel DOC^TM^ EZ Imager (Bio-Rad Laboratories Inc.).

### 2.8. Western Blot Analysis

The presence or absence of immunogenic proteins was qualitatively analyzed using Western blotting. SDS-PAGE was performed as described in Section 2.7 on samples of the culture medium incubated at 35 °C. However, a different protein marker (model: 1610375; Bio-Rad Laboratories Inc.) was used. The protein bands were transferred onto Immobilon^TM^ PVDF membranes (Merck KGaA, Darmstadt, Germany) using the Trans-Blot Turbo Transfer System (Bio-Rad Laboratories Inc.). The membranes were blocked with 5% skim milk for 1 h and then incubated overnight with an anti-gliadin antibody [14D5] (ab36729, Abcam Inc., Cambridge, UK) diluted in Tris-buffered saline tween 20 (TBST; 1:1000). According to the product datasheet of the Abcam company, the anti-gliadin antibody reacts specifically with the deaminated peptide (KLQPFPQPELPYPQPQ), which comprises the 33 mer sequence part of the wheat gliadin (residues 56–88). After washing thrice with TBST, the membranes were probed with a secondary antibody for 2 h and washed with TBST. Immunodetection was performed using Hyperfilm and ECL reagents (Amersham-GE Healthcare Europe GmbH, Glattbrugg, Switzerland), and images were acquired using the Chemiluminescence Imaging System (UVITEC Ltd., Cambridge, UK).

### 2.9. Measurement of Crude Enzyme Activity at Different pH

The pH range for the activation of the crude enzyme and the effect of gluten or gliadin present in the culture medium on the enzyme activity was determined. The bacterium was inoculated at the final cell density of approximately 6 log CFU/mL in TSB supplemented with 0.1–0.2% gluten or gliadin and incubated at 35 °C for 15 h. DCZ of the ten-fold concentrated cell-free culture (as a crude enzyme) was determined using 0.5% gluten plus 1.5% agar plates. The pH of the agar plates was adjusted from 3 to 8 using HCl or NaOH. The DCZ of the crude enzyme was analyzed as described in Section 2.4.

### 2.10. Statistical Analysis

Data of bacterial cell counts, culture medium pH, DCZ of the crude enzyme, and biofilm formation at 25, 35, and 45 °C for 48 h were statistically assessed using the SPSS program (IBM, Armonk, NY, NJ, USA). Data are expressed as the mean ± standard deviation (SD), and a significant difference (*p* < 0.05) in the means was evaluated using Tukey’s test.

## 3. Results and Discussions

### 3.1. Isolation and Morphological Analysis of the Gluten-Degrading Bacterium

To isolate gluten-degrading bacterium, wheat grain powder was fermented in TSB (30 °C, 12 h) and then spread on TSA containing 0.5% gluten. Multiple colonies grew and one colony that formed a clear zone was collected. The colony was validated by streaking on TSA containing 0.5% gluten and successfully pure-cultured using an agar medium supplemented with 0.5% gluten and 0.1% glucose (Figure 1). The clear zone indicated the decomposition of gluten proteins. The colony on TSA containing 0.5% gluten had a sticky surface and was creamish in color, flat, and non-glossy. The bacterium observed using SEM was *Bacilli*-shaped with an approximate diameter of 0.58 µm and length of 2–5 µm.

### 3.2. Species Identification of the Bacterium

16S rDNA sequencing was used to identify the species of the isolated bacterium. The results revealed 99% similarity between the isolated bacterium and *Bacillus amyloliquefaciens* subsp. *plantarum* (Figure 2). The bootstrap value of the phylogenetic tree was obtained using 1000 replicates, and the value of *B. amyloliquefaciens* subsp. *plantarum* CP000560 was 64%, which is similar to *B. methylotrophics* and *B. siamensis* more than *B. amyloliquefaciens*. The bacterium was deposited in the KCCM, an international depository authority for microorganisms (accession number: KCCM 12688P).

*B. amyloliquefaciens* is an endo-spore-forming and Gram-positive bacterium. It is ubiquitous and usually found in foods, plants, soils, and water [26]. The bacterium contributes to hydrolyze different plant and animal products; synthesizes different enzymes, proteins, and carbohydrates; and is utilized for prebiotic and probiotic purposes, making it more valuable in food processing [27]. The type strains of *B. amyloliquefaciens* DSM7T and *B. amyloliquefaciens* subsp. *plantarum* FZB42T have a common 3345 coding sequence (CDS) in their core genomes, but they have different unique CDS [26]. The gluten-degrading activity of *B. amyloliquefaciens* species has been mentioned in study [27], but there was no report about gluten-degrading ability of *Bacillus amyloliquefaciens* subsp. *plantarum.*

### 3.3. Culture Conditions for Gluten-Degrading Enzyme Production

The culture conditions of the *B. amyloliquefaciens* subsp. *plantarum* KCCM 12688P were investigated to consider utilization of extracellular gluten-degrading enzymes. Figure 3 presents changes in the bacterial cell count, culture medium pH, bacterial enzyme activity, and biofilm formation according to culture temperature and time. The initial cell count was approximately 5.9 log CFU mL^−1^ (Figure 3A). The maximal cell count, which was approximately 7.3 log CFU mL^−1^, was similar in all the culture temperatures (25, 35, and 45 °C). However, the reaching time was slightly different; it was reached at 12 h in both 35 °C and 45 °C, but the maximal cell count at 25 °C was reached at 24 h. Nutrients and culture temperature affect the time required for cell growth to reach the maximum [28]; the maximal cell count of *B. amyloliquefaciens* is known to be reached at 11–12 h at 37 °C in the optimal medium [29]. The pH value of the culture medium at 25 °C was maintained for 48 h, while that of the culture medium at 35 °C was temporarily decreased at 9 h (Figure 3B). The pH value of the culture medium at 45 °C significantly decreased for 12 h (*p* < 0.05) and was retained until 36 h; after that, it increased.

To evaluate gluten-degrading enzyme production, ten-fold concentrated cell-free cultures were used as the crude enzyme. The enzyme activity was different depending on the culture temperatures: change in activity at 25 °C between 12 and 24 h; at 35 °C between 9 and 18 h; and at 45 °C between 6 and 12 h, respectively (Figure 3C). We found that the culture times for enzyme production were gradually reduced with increasing culture temperature. Interestingly, the DCZ of the crude enzyme isolated at 45 °C was the highest at 9 h, but it was decreased at 12 h and was not detected at 18 h. At this time, the pH value of the culture medium was notably decreased and bacterial biofilm formation was significantly enhanced, compared to those at 25 °C and 35 °C (Figure 3D). Biofilm can be defined as a community of microorganisms attached to a surface, which are generally encapsulated and protected by an extracellular matrix composed of various biopolymers. As the biofilm reaches maturity, the bacteria enter a stationary phase, making it difficult to produce enzymes [24]. In general, biofilm formation is influenced by growth temperature, pH condition, starvation, and metabolite accumulation to protect against various stresses [30,31].

### 3.4. Bacterial Carbohydrate Fermentation and Enzyme Activity

Bacterial carbohydrate fermentation and enzyme activity were screened to estimate medium composition for bacterial growth and further utilization. The results showed that *B. amyloliquefaciens* subsp. *plantarum* KCCM 12688P fermented D-glucose, D-fructose, D-mannose, esculin ferric citrate, D-cellobiose, and sucrose, and produced esterase (C4), esterase lipase (C8), acid and alkaline phosphatases, leucine arylamidase, α-chymotrypsin, naphthol-AS-BI-phosphohydrolase, and α-glucosidase.

### 3.5. Decomposition Patterns of Gluten Proteins

Figure 4 illustrates decomposition patterns of gluten proteins in a culture medium at different temperatures and times. The gluten used as a standard was composed of high molecular weight (MW) glutenin (80–170 kDa), ω-gliadin (30–75 kDa), and α-, β-, and γ-gliadins (30–45 kDa), while standard gliadin was mostly composed of α-, β-, and γ-gliadin [9]. The SDS-PAGE analysis revealed decomposition of gluten proteins by *B. amyloliquefaciens* subsp. *plantarum* KCCM 12688P at all culture temperatures. However, times at which decomposition occurred were dependent on culture temperatures. The band intensities of glutenin and gliadin proteins of the culture medium at 25 °C decreased at 48 h. On the contrary, glutenin and gliadin proteins of culture media at 35 °C and 45 °C began to decompose at 24 h, but the decomposition was obvious at 48 h. In particular, high-MW glutenin proteins were decomposed first, followed by gliadin proteins, regardless of the culture temperature. In the results of growth conditions for enzyme production, the culture temperature was suggested at 25 °C and 35 °C, because of the excess biofilm formation in the culture medium at 45 °C, which can make it difficult to isolate the enzymes from the culture medium. However, to use microorganisms directly, 35 °C or 45 °C is better for the decomposition of gluten proteins.

Wheat gluten is composed of glutenins and gliadins, and the latter have different classes of protein fractions including α-, β-, γ-, and ω-gliadins [3,7]. Among these proteins, some hydrolyzed peptide residues from α-gliadin or γ-gliadin are known to activate T cells, associated with gluten intolerance [9,11,32]. In this study, the intensities of gluten protein bands estimating as α- and γ-gliadin sites, including the protein bands of high- and low-MW glutenins and other gliadins, were reduced by *B. amyloliquefaciens* subsp. *plantarum* KCCM12688P at 35 °C and 45 °C at 48 h.

### 3.6. The Presence of Immunogenic Protein

The presence of immunogenic proteins was verified using Western blotting with an anti-gliadin antibody (Figure 5), which reacts specifically with the deaminated peptide (KLQPFPQPELPYPQPQ), comprising the 33 mer sequence part of the wheat gliadin (residues 56–88) [33]. The immunogenic reaction of gluten and gliadin proteins was shown in the range of 25–50 kDa. This aspect was accompanied with the study of [33]. Consistent with the results of SDS-PAGE, the immunogenic proteins related to α/β-gliadins were markedly decomposed by *B. amyloliquefaciens* subsp. *plantarum* KCCM 12688P at 35 °C from 24 h.

The immunoblotting was conducted to provide a more precise qualification of immunogenic protein related to α/β-gliadin and their hydrolysis degree by *B. amyloliquefaciens* subsp. *plantarum* KCCM12688P. The α- and β-gliadins have similar structures and sequences (250 and 300 amino acid residues), and these are characterized by high proline and glutamine contents as with a similar motif (PQQPFPQQ) [34]. In particular, α-gliadins consist of five residues and a central domain that contains the most characteristic immunogenic sequence 33 mer peptides, comprising six overlapping epitopes regarded as a trigger of celiac disease pathogenesis [35]. Although the analysis to demonstrate the decomposition of 33 mer peptides alone from α-gliadins has not been conducted, in a previous study, decomposition of the 33 mer immunogenic peptide by *B. subtilis/amyloliquefaciens* was reported [36].

### 3.7. pH Range for Crude Enzyme Activation and the Effect of Gluten Protein Additions on the Enzyme Activity

To investigate the active pH range of the crude enzyme, DCZ of the concentrated cell-free culture (as the crude enzyme) was evaluated using 0.5% gluten agar adjusted at a pH range of 3 to 8. In addition, 0.1% and 0.2% gluten or gliadin were added to the culture medium to analyze their effect on enzyme production by the bacterium (Figure 6). The crude enzyme did not act at pH 3, while it decomposed gluten at pH 5, 7, and 8. Furthermore, the enzyme activities were significantly increased in the culture medium with 0.2% gliadin (*p* < 0.05).

It was reported that several serine-type enzymes produced by *Aspergillus niger*, *Sphaerobacter thermophiles*, and *Sphingomonas capsulate* degrade gluten proteins [18], but the active pH was different according to the species *Aspergillus niger*, pH 4–5; *Sphaerobacter thermophiles*, pH 6–8; and *Sphingomonas capsulate*, pH 7–8 [8,13].

## 4. Conclusions

The aim of this study was to isolate gluten-degrading bacterium and establish suitable conditions for their growth and enzyme production. The gluten-degrading bacterium isolated from wheat grains was identified as *Bacillus amyloliquefaciens* subsp. *plantarum* KCCM 12688P. The gluten-degrading bacterium grew rapidly at 45 °C, at which the production of the gluten-degrading enzyme was faster than the other temperatures, but the biofilm was excessively formed, which may make it difficult to isolate the enzyme. Gluten proteins composed of glutenins and α-, β-, γ-, and ω-gliadins were reduced by the bacterium, and the immunogenic proteins related to α/β-gliadins were markedly reduced. The crude extracellular enzymes with gluten-degrading function acted in the pH ranges of 3 to 8, and the gliadin addition into the culture medium enhanced enzyme production of the bacterium. Therefore, the gluten-degrading enzyme can be used in the baking and beer industries to reduce immunogenic proteins in foods containing gluten.

If further studies for purification of the enzyme specific to the immunogenic protein and its characteristics are conducted, it may contribute to the treatment of individuals with gluten intolerance using the gluten-degrading enzyme produced by *B. amyloliquefaciens* subsp. *plantarum* KCCM 12688P.

## Figures and Tables

**Figure 1 microorganisms-11-02884-f001:**
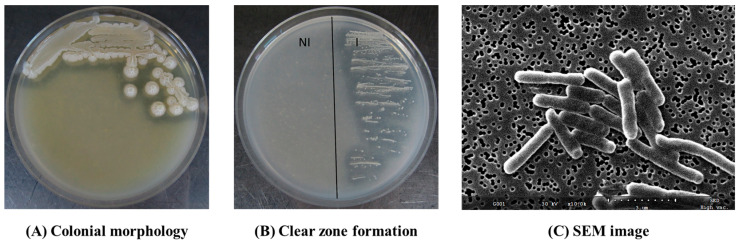
The images of colony (**A**) and clear zone (**B**) formed by the bacterium isolated from wheat grains on gluten-containing agars, and scanning electron microscope (SEM) observation (**C**). The bacterium was streaked on tryptic soy agar (TSA) containing 0.5% gluten (**A**) or 0.5% gluten plus 0.1% glucose agar (**B**). NI: No inoculation. I: inoculation. SEM image was obtained at ×10,000 magnification.

**Figure 2 microorganisms-11-02884-f002:**
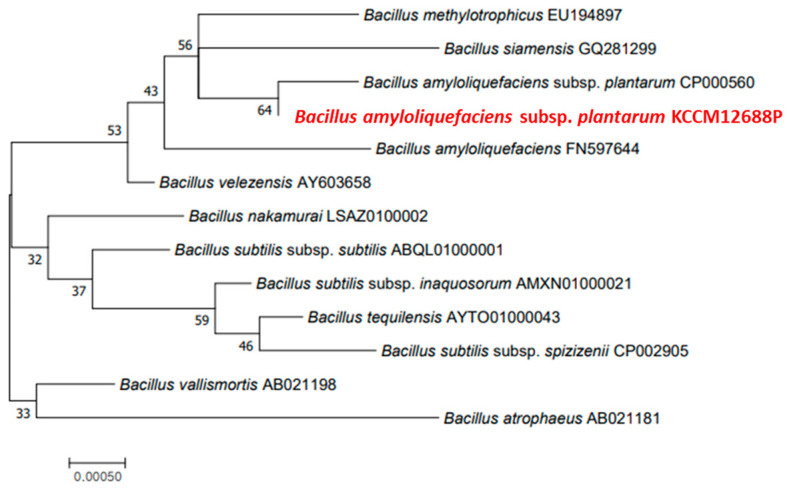
Phylogenetic tree of *B. amyloliquefaciens* subsp. *plantarum* KCCM 12688P and related taxa based on 16S rDNA gene sequences. The numbers represent the bootstrap values, which were obtained from 1000 repeats.

**Figure 3 microorganisms-11-02884-f003:**
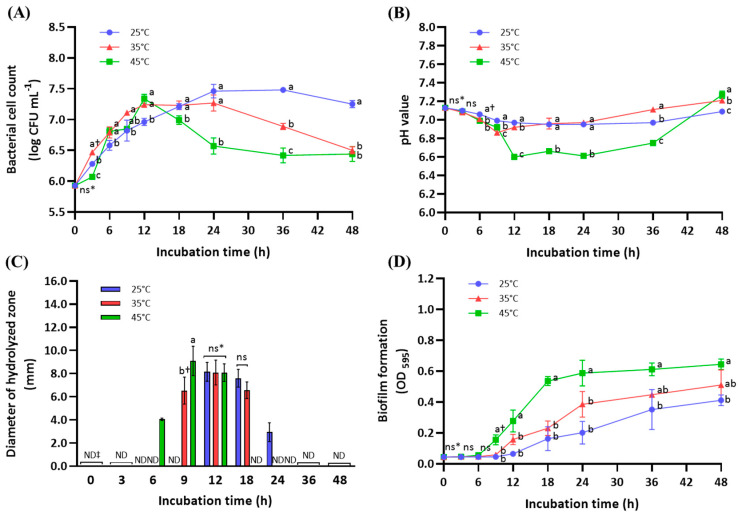
The change in bacterial cell count (**A**), culture medium pH (**B**), crude enzyme activity (**C**), and biofilm formation (**D**) in tryptic soy broth (TSB) inoculated with *B. amyloliquefaciens* subsp. *plantarum* KCCM 12688P. * No significant difference (*p* < 0.05). ^†^ Different letters indicate significant differences between incubation times (*p* < 0.05). ^‡^ No decomposition.

**Figure 4 microorganisms-11-02884-f004:**
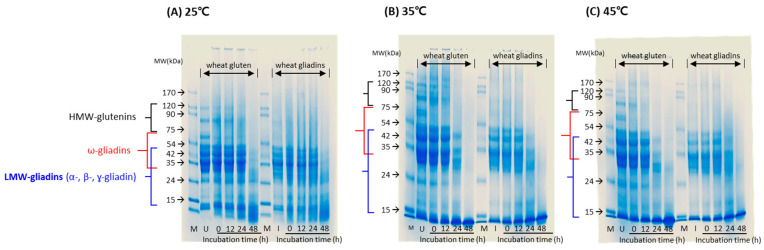
Decomposition patterns of gluten proteins in culture medium inoculated with *B. amyloliquefaciens* subsp. *plantarum* KCCM 12688P at 25 °C (**A**), 35 °C (**B**), and 45 °C (**C**). M, protein marker; U, wheat gluten standard; I, wheat gliadin standard.

**Figure 5 microorganisms-11-02884-f005:**
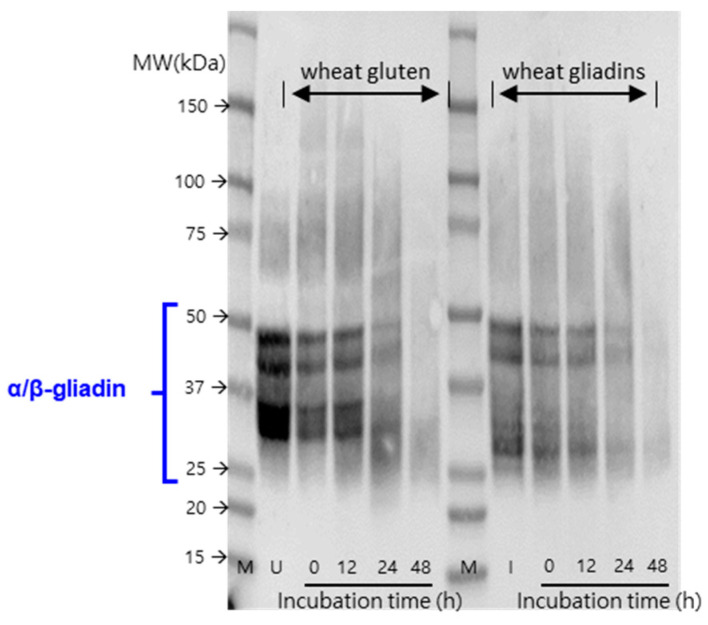
The presence of immunogenic protein in the culture medium (containing gluten or gliadin) inoculated with *B. amyloliquefaciens* subsp. *plantarum* KCCM 12688P at 35 °C. α/β-gliadin was used as antibody. M, protein marker; U, wheat gluten standard; I, wheat gliadin standard.

**Figure 6 microorganisms-11-02884-f006:**
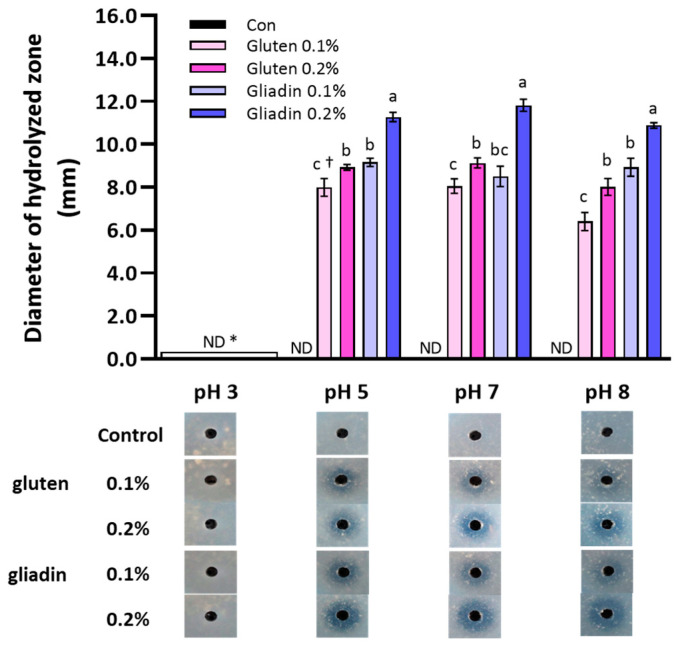
The active pH range of the crude enzyme and the effect of gluten protein additions into the culture medium on enzyme activity. Gluten-degrading activity of the crude enzyme was evaluated using 0.5% gluten plus 1.5% agar plates adjusted to desired pH using HCl or NaOH. The cell-free culture used was ten-fold-concentrated as the crude enzyme. The culture groups were as follows: control, TSB without the bacterium and gluten proteins; S1, TSB with the bacterium and 0.1% gluten; S2, TSB with the bacterium and 0.2% gluten; S3, TSB with the bacterium and 0.1% gliadin; S4, TSB with the bacterium and 0.2% gliadin. * No significant difference (*p* < 0.05). ^†^ Different letters indicate significant differences between incubation times (*p* < 0.05).

## Data Availability

Data presented in this study are available in the article.

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
