# Peer review of "Identification and Growth Characteristics of a Gluten-Degrading Bacterium from Wheat Grains for Gluten-Degrading Enzyme Production"

_microorganisms, 2023, doi:10.3390/microorganisms11122884_

Round 1
Reviewer 1 Report
Comments and Suggestions for Authors
This manuscript reports on the isolation and description of of a bacterium from wheat grains, identified as Bacillus amyloliquefaciens subsp. plantarum. The bacterium secretes gluten-degrading activity both on glutenin and gliadin proteins of 20-45 kDa identified by anti-gliadin antibodies.
The characterization of the degradative activity is very preliminary without any significant purification or molecular identification. It is suggested that the unidentified protease responsible for the activity could be of interest for biotechnological applications related to gluten sensitivity. However, it is not known whether the activity corresponds to one or several enzymes.
There is no evidence that smaller immunogenic peptides (26-33 amino acids) are actually degraded.
MAJOR ISSUES
1.- As recognized by the authors (lines 394-396), the enzyme(s) responsible for gluten degradation need some characterization. Separation techniques should be applied to obtain at least a partially purified preparation with a defined molecular weight, and to estimate possible heterogeneity.
2.- The ability to degrade short immunogenic peptides (26-mer, 33-mer) should be demonstrated. In the absence of this information, the potential use for reducing allergens in food materials containing gluten remains strongly speculative.
3.- Biofilm formation B. amyloliquefaciens subsp. plantarum KCCM 12688P is stated (lines 273 and 359-360) but not shown in Results and in Methods.
MINOR ISSUES
References must be numbered in order of appearance in the text.
There are at least two published reports of B. amyloliquefaciens activities on gluten, which are not cited in the manuscript:
https://doi.org/10.22059/jfabe.2022.338961.1110
https://dx.doi.org/10.1021/acs.jafc.0c06396
Lines 12-14. It is stated that: “There are numerous cures and treatments for gluten intolerance, but a gluten-free diet and use of gluten-degrading enzymes are promising strategies for its prevention”. Gluten-free diet is the only well establshed treatment for gluten intolerance. Saying that it is a “promising strategy” is not in agreement with its prominent status in treatment.
Line 18. Change “activation” to “activity”.
Line 19. Is there any reason to say that “enzymes” (plural) are being produced?
Line 77. Change “activate” to “active”.
Lines 150, 205, 288. References should be cited by number and included in the list.
Line 262. What is meant by “reaching time”?
Figure 4 and Figure 5. What do the arrows indicate?
Line 350. Delete “Nonetheless”.
Lines 368-370. The authors mention that “A previous study demonstrated that α-gliadin degraded by B. amyloliquefaciens subsp. plantarum KCCM 12688P contains 33-mer immunogenic peptides”. The meaning of this comment is unclear: does it mean that undegraded gliadin contains these peptides, or that after gliadin degradation, the immunogenic peptides remain intact?
Comments on the Quality of English LanguageMinor edits are required
Author Response
Dear Reviwer 1,
We deeply appreciate your valuable comments, and thanks to your sharp criticisms and correct comments, we have been able to better understand the important issues of the manuscript. The manuscript has been extensively revised according to your suggestion. To enhance understating of the results, the discussion section has been combined with the results. Two data has been supplemented in the Figures 3 (biofilm formation) and 6 (diameter of the hydrolyzed zone). We provide a point-by-point response to the comments; All changes are highlighted in red. We would like to thank you again for your criticisms.
*The answers have been attached as a file.

Reviewer 2 Report
Comments and Suggestions for Authors
Celiac disease is a chronic autoimmune disease caused by the ingestion of gluten by a genetically susceptible individual. A gluten-free diet is still the only recommended method of treating celiac disease and usually leads to an improvement or disappearance of enteropathy and symptoms. However, the possibility of unintentional ingestion of gluten, which causes chronic changes in the integrity of the mucous membrane and persistent symptoms, requires the development of targeted pharmacological therapy in addition to a gluten-free diet. The purpose of this article is to find available additional therapies for the treatment of celiac disease. In this regard, the presented work on the search for new sources of enzymes that can effectively hydrolyze gluten may be of some interest. However, there are many questions about the presented article. It is replete with a lot of inaccuracies and errors and cannot be published without a serious revision. Perhaps they are connected with insufficiently accurate representation of the drawings, leading to a discrepancy between the results described in the text and those presented in the drawings.
Main remarks:
Line 10 – does not contain immunogenic peptides, but forms during hydrolysis
Line 14 – gluten-free diet and gluten-degrading enzymes are completely different things, if the first is generally accepted and widely used, then the second is only at the development stage
Line 37 – explain that we are talking about undigested proteins and the resulting immunogenic peptides
Line 45 - do not contain peptides, but form them during hydrolysis
Line 51 - replace "recommended" with considered
Line 63 - prolamines is the common name of the alcohol-soluble fraction. It should be written that immunogenic peptides are formed during the hydrolysis of proline/glutamine-rich prolamines
Line 64 - subtilisin is a postglutamine-cleavage enzyme rather than a postproline-cleavage
Line 65 - there is, apparently, a reference to prolyl carboxypeptidase
Line 83 – water for what? for washing? Explain
Line 261 – according to Fig. 3A the similarity is observed not at 48 h, but only up to 18 h.
Line 273 - DCZ was highest at 45 °C and 9 h, when the pH of the medium decreased slightly. At 12 h DCZ was the same at all temperatures (Fig. 3C)
Line 278 - the meaning of lowercase is not clear and requires explanation
Line 297 - why is it concluded that 25°C and 35°C are optimal for enzyme production?
Line 305 - immunogenic peptides are not α/β-gliadin subunits. By molecular weight, they are significantly lower than the lower limit of the taps used. Therefore, it is not correct to draw a conclusion about their destruction. In Figure 4A (24-48 hours) we just see the accumulation of such low-molecular components
Line 310 – this cannot be said from the presented electropherograms. We see the disappearance, perhaps, of α/β-gliadin subunits, but not the complete destruction of immunogenic peptides, whose molecular weights may be 1-3 kDa.
Line 326 – it would be interesting to find out to what extent the increase in activity is associated with the addition of gliadin to the medium, and does not depend simply on the addition of any protein to the medium, for example, casein
Line 342 – here it is necessary to write not about immunogenic peptides, but simply about the removal of gluten
Line 344 – what "formulated gluten" are we talking about? Peptides are not additives in gluten, but what is obtained from it during hydrolysis
Line 359 – where does this come from? Figure 3A does not show this. Up to 18 hours everything is about the same
Line 364 – more precisely, these are their peptides obtained by hydrolysis
Line 366 – not containing peptides, but forming them during hydrolysis
Line 369 - does not contain a peptide, but forms during hydrolysis
Line 372 – peptides are not in gliadin, but gliadin peptides
Lines 374-375 – replace with "immunogenic gluten peptides"
Line 377 – immunogenic gluten peptides, not subunits
Line 381 - there is no visual difference between pH 5, 7 and 8 in Fig. 6
Line 386 – add “isolate and purify enzyme”
Line 406 – gliadin does not consist of peptides, they are formed during its hydrolysis. Based on the presented results, we can only talk about the presumed degradation of gliadin (gluten), but not its immunogenic peptides
Line 408 – a more precise determination of the optimum pH is required here, since the results given do not make it possible to draw such a conclusion
Minor notes:
From line 32 onwards in the text - usually in MDPI journals, links are in citation order
Line 228 – (A) in bold
Line 231 – instead of (a) you need (A) and in bold
Line 287 – remove the comma and the author's link
Lines 276-279 - in the caption to Fig. 3, capitalize the letters indicating a particular picture
Lines 302-304 - in the caption to Fig. 4, capitalize the letters indicating a particular picture
Line 375 – replace the author’s link with a digital one
Line 443 - doi do not match the article
Author Response
Dear Reviewer 2,
We deeply appreciate your valuable comments, and thanks to your sharp criticisms and correct comments, we have been able to better understand the important issues of the manuscript. The manuscript has been extensively revised according to your suggestion. To enhance understating of the results, the discussion section has been combined with the results. Two data has been supplemented in the Figures 3 (biofilm formation) and 6 (diameter of the hydrolyzed zone). We provide a point-by-point response to the comments; All changes are highlighted in red. We would like to thank you again for your criticisms.
*The answers have been attached as a file.

Reviewer 3 Report
Comments and Suggestions for Authors
Wheat gluten contains immunogenic peptides that are difficult to digest by gastrointestinal proteases and negatively affect immune responses in humans. Gluten intolerance is a problem in countries where wheat is a staple food. There are numerous cures and treatments for gluten intolerance, but a gluten-free diet and use of gluten-degrading enzymes are promising strategies for its prevention. Here, we isolated a gluten-degrading bacterium, Bacillus amyloliquefaciens subsp. plantarum, from wheat grains. The culture conditions for enzyme production or microbial use were optimized based on gluten decomposition patterns. Additionally, the pH range for the activation of the crude enzyme was investigated. The bacterial production of gluten-degrading enzymes was temperature-dependent, and the culture time decreased with increasing culture temperature. The bacterium decomposed high molecular weight glutenin proteins first, followed by gliadin proteins, regardless of the culture temperature. Western blotting revealed that the bacterium decomposed immunogenic α/β-gliadin subunits. The crude enzyme was active in the pH range 5–8, and enzyme production was increased by adding gliadin. Collectively, B. amyloliquefaciens subsp. plantarum can be used to produce gluten-degrading enzymes to prevent gluten intolerance.
Overall, this is a worthy paper containing interesting results, but the summary of this paper is relatively shallow. The views expressed are not clear enough, and the writing of the whole article is not rigorous enough. We suggest that this manuscript should be greatly revised.
“1. Introduction”:
Line 40-56 This description of gluten allergy in wheat should be introduced. Please refer this reference (Journal of Food Processing and Preservation,2021, 45(9), e15684.).
The peptide about gluten has different function. Please refer this reference (Neuroscience Letters, 2016, 631(19): 30-35).
2.4. Analysis of bacterial carbohydrate fermentation and enzyme activity
It should have the optimized design experiment. Please refer this reference (Food Chemistry, 402(2023): 134231.).
3.5. Decomposition patterns of gluten proteins and the presence of immunogenic peptide
Please give us the reason about immunogenic peptide.
Figure and Table
It is suggested to rearrange the Figure to make the layout clearer.
Please update the reference.
Comments on the Quality of English LanguageWheat gluten contains immunogenic peptides that are difficult to digest by gastrointestinal proteases and negatively affect immune responses in humans. Gluten intolerance is a problem in countries where wheat is a staple food. There are numerous cures and treatments for gluten intolerance, but a gluten-free diet and use of gluten-degrading enzymes are promising strategies for its prevention. Here, we isolated a gluten-degrading bacterium, Bacillus amyloliquefaciens subsp. plantarum, from wheat grains. The culture conditions for enzyme production or microbial use were optimized based on gluten decomposition patterns. Additionally, the pH range for the activation of the crude enzyme was investigated. The bacterial production of gluten-degrading enzymes was temperature-dependent, and the culture time decreased with increasing culture temperature. The bacterium decomposed high molecular weight glutenin proteins first, followed by gliadin proteins, regardless of the culture temperature. Western blotting revealed that the bacterium decomposed immunogenic α/β-gliadin subunits. The crude enzyme was active in the pH range 5–8, and enzyme production was increased by adding gliadin. Collectively, B. amyloliquefaciens subsp. plantarum can be used to produce gluten-degrading enzymes to prevent gluten intolerance.
Overall, this is a worthy paper containing interesting results, but the summary of this paper is relatively shallow. The views expressed are not clear enough, and the writing of the whole article is not rigorous enough. We suggest that this manuscript should be greatly revised.
“1. Introduction”:
Line 40-56 This description of gluten allergy in wheat should be introduced. Please refer this reference (Journal of Food Processing and Preservation,2021, 45(9), e15684.).
The peptide about gluten has different function. Please refer this reference (Neuroscience Letters, 2016, 631(19): 30-35).
2.4. Analysis of bacterial carbohydrate fermentation and enzyme activity
It should have the optimized design experiment. Please refer this reference (Food Chemistry, 402(2023): 134231.).
3.5. Decomposition patterns of gluten proteins and the presence of immunogenic peptide
Please give us the reason about immunogenic peptide.
Figure and Table
It is suggested to rearrange the Figure to make the layout clearer.
Please update the reference.
Author Response
Dear Reviewer 3,
We deeply appreciate your valuable comments, and thanks to your sharp criticisms and correct comments, we have been able to better understand the important issues of the manuscript. The manuscript has been extensively revised according to your suggestion. To enhance understating of the results, the discussion section has been combined with the results. Two data has been supplemented in the Figures 3 (biofilm formation) and 6 (diameter of the hydrolyzed zone). We provide a point-by-point response to the comments; All changes are highlighted in red. We would like to thank you again for your criticisms.
*The answers have been attached as a file

Reviewer 4 Report
Comments and Suggestions for Authors
Reviewer comments:
The authors have presented their work in a manuscript entitled “Identification and growth characteristics of a gluten-degrading bacterium from wheat grains for gluten-degrading enzyme production.” However, the manuscript cannot be accepted for publishing in ‘Microorganisms’ journal for the following reasons:
1. The manuscript lacks novelty and I didn’t find the significance of the research work. This seems to be a preliminary study and needs more experiments/data to be accepted as a full-length paper. Moreover, many other studies reported attempting to use gluten-degrading enzymes from Bacillus species.
2. The manuscript lacks the development of a bioprocess and is presented with preliminary data.
3. The introduction section is not appealing and is too general not sticking to the main theme of this research.
4. The manuscript lacks a proper discussion and interpretation of the data.
5.
Author Response
Dear reviewer 4,
We deeply appreciate your valuable comments, and thanks to your sharp criticisms and correct comments, we have been able to better understand the important issues of the manuscript. The manuscript has been extensively revised according to your suggestion. To enhance understating of the results, the discussion section has been combined with the results. Two data has been supplemented in the Figures 3 (biofilm formation) and 6 (diameter of the hydrolyzed zone). We provide a point-by-point response to the comments; All changes are highlighted in red. We would like to thank you again for your criticisms.
*The answers have been attached as a file.

Round 2
Reviewer 1 Report
Comments and Suggestions for Authors
In the revised manuscript, out of the three major points raised by this reviewer only one (point 3) has been addressed as requested, by showing data not shown in the original version. Major points 1 and 2 (which are very important to modify the “preliminary” character of the study) have not been addressed. In their cover letter, the authors just state their plans to address them in future work.
The current manuscript just shown the presence of gluten degradative activity that may correspond to more than one enzyme. Some purification and characterization of the enzyme(s) involved is necessary. The preliminary character of the manuscript is reinforced by the occurrence of three published reports of B. amyloliquefaciens activities on gluten: two are references #26 (lines 284-285) and #35 (line 404), another (not cited) is “Functional properties and emulsion stability of wheat gluten hydrolysates produced by endopeptidases from Bacillus licheniformis and Bacillus amyloliquefaciens” (https://doi.org/10.22059/jfabe.2022.338961.1110).
In the absence of evidence that smaller immunogenic peptides (26-33 amino acids) are actually degraded by the B. amyloliquefaciens activities, their potential use for reducing allergens in food materials containing gluten remains strongly speculative.
Comments on the Quality of English Languageminor edits are needed
Author Response
Dear Reviewer 1,
We are grateful to the reviewer for the valuable comments.
The isolation and identification of effective microorganisms from raw materials are important, and the characterization of the bacterial growth for producing desirable enzymes is also important. The literature suggested by the reviewer address the commercial enzymes, it excludes the characteristics of the microorganisms.
As you well known, even the same species of bacteria, they frequently produce different enzymes which are depended on the strain. In this regard, we think that the reviewer mentioned that the characterization of the specific enzyme can reinforce this manuscript.
However, the KCCM12688P strain is closer to B. methylotrophicus and B. siamensis than B. amyloliquefaciens, and there are no reports of gluten degrading bacterium for this subspecies.
It is difficult to say that the crude enzyme is a single enzyme, from the result of the western blotting, we can know that there is an enzyme that breaks down the immunogenic protein. In other words, specifying the enzyme by a specific substrate degradation test is not much different to western blot.
The growth characteristics of the bacterium for crude enzyme production should be investigated before step of isolation of the specific enzyme, and the enzyme characterization requires some purification process, which spend a lot of time and effort.
Regarding the immunogenic peptides, as we dscribed in the manuscript, anti-gliadin antibody reacts specifically with the deaminated peptide (KLQPFPQPELPYPQPQ), comprising the 33-mer sequence part of the wheat gliadin (residues 56-88). Although the analysis to demonstrate the decomposition of 33-mer peptides alone from α-gliadins has not been conducted, in other report, they interprept the immunoblotting results similary [reference 33].
Reviewer 2 Report
Comments and Suggestions for Authors
The authors have done quite a lot of work, eliminating the mistakes made and supplementing the article with new experimental data. However, the article requires further improvement.
Remarks:
Line 33 – it is not necessary to insert "immunogenic peptides" into keywords, since peptides are not discussed in the article
Lines 40-44 – the sentence is not clear. It can be replaced with "Some gluten proteins form immunogenic peptides during hydrolysis, which are poorly digested by the proteases of the gastrointestinal tract and can cause diseases in sensitive people, such as ....."
Line 94 – it remains unclear why wheat grains that need to be ground are first soaked and then dried.
Lines 311-313 – the proposal to change, writing that "changes in activity at 25o between 12 and 24 hours, at 35o – between 9 and 18 hours, аnd at 45o – between 6 and 12 h."
Line 317 – remove "even if" and insert on the same line "pH of the culture medium"
Line 323 – "phase" instead of "phage"
Line 378 – there is no data on immunogenic peptides, you can only write about immunogenic proteins
Fig. 5 – in the caption, indicate that this is Western blotting and that the results were obtained at two temperatures
Lines 426-433 – it is not clear why the paragraph was added, which only talks about prolylendopeptidase. It can only be considered as one of the enzymes capable of breaking down gluten. There is no evidence in the article that the detected enzymatic activity may belong to prolylendopeptidase
Lines 454-455 – here we can only talk about an enzyme capable of breaking down gluten proteins, but not about an enzyme specific to immunogenic peptides
Line 558 – doi does not match the given link. The article is in the public domain with doi: 10.3390/pharmaceutics13101603. Apparently, it makes sense to check the compliance of the doi with the given links from other articles as well
Author Response
Dear Reviewer 2,
We are grateful to the reviewer for the valuable comments. We believe your comments will greatly improve the quality of our manuscript.
R2: The authors have done quite a lot of work, eliminating the mistakes made and supplementing the article with new experimental data. However, the article requires further improvement.
A: We carefully revised our manuscript with your recommendation.
R2: Line 33 – it is not necessary to insert "immunogenic peptides" into keywords, since peptides are not discussed in the article.
A: "immunogenic α/β-gliadin peptides" has been revised by ‘α/β-gliadin protein’ on the line 33.
R2: Lines 40-44 – the sentence is not clear. It can be replaced with "Some gluten proteins form immunogenic peptides during hydrolysis, which are poorly digested by the proteases of the gastrointestinal tract and can cause diseases in sensitive people, such as ....."
A: As reviewer's suggestion, the sentence has been changed on the lines 40-43.
R2: Line 94 – it remains unclear why wheat grains that need to be ground are first soaked and then dried.
A: ‘dry’ has been replaced to ‘allowed to drain’ on the lines 94-95. For eliminate pericarp, swelling and water drain processes are required.
R2: Lines 311-313 – the proposal to change, writing that "changes in activity at 25o between 12 and 24 hours, at 35o – between 9 and 18 hours, аnd at 45o – between 6 and 12 h."
A: As reviewer's recommendation, the sentence has been changed on the lines 313-315.
R2: Line 317 – remove "even if" and insert on the same line "pH of the culture medium"
A: As reviewer's recommendation, "even if" has been deleted, and "the pH value" has been also changed to "the pH of the culture medium".
R2: Line 323 – "phase" instead of "phage"
A: It has been corrected by ‘phase’ on the line 325.
R2: Line 378 – there is no data on immunogenic peptides, you can only write about immunogenic proteins.
A: The "peptides" has been changed to "proteins" on the line 380.
R2: Fig. 5 – in the caption, indicate that this is Western blotting and that the results were obtained at two temperatures.
A: As reviewer's comment, it has been corrected on the lines 389 and 393.
R2: Lines 426-433 – it is not clear why the paragraph was added, which only talks about prolylendopeptidase. It can only be considered as one of the enzymes capable of breaking down gluten. There is no evidence in the article that the detected enzymatic activity may belong to prolylendopeptidase.
A: It has been changed to “ It has been reported that several serine type enzymes produced by Aspergillus niger, Sphaerobacter thermophiles, and Sphingomonas capsulate degrade gluten proteins” on the lines 428-230.
R2: Lines 454-455 – here we can only talk about an enzyme capable of breaking down gluten proteins, but not about an enzyme specific to immunogenic peptides.
A: It has been changed to “ If further studies for purification of the enzyme specific to the immunogenic protein and its characteristics are conducted” on the lines 453-454.
R2: Line 558 – doi does not match the given link. The article is in the public domain with doi: 10.3390/pharmaceutics13101603. Apparently, it makes sense to check the compliance of the doi with the given links from other articles as well.
A: It has been corrected.
Thank you for your consideration.
Reviewer 3 Report
Comments and Suggestions for Authors
The author did not responded to the reviewer's comment point by point. The reference should be updated in recent years.
Comments on the Quality of English LanguageThe author did not responded to the reviewer's comment point by point. The reference should be updated in recent years.
Author Response
Dear Reviewer 3,
We are grateful to the reviewer for the valuable comment. We believe your comment will greatly improve the quality of our manuscript.
R3: The reference should be updated in recent years.
A: A reference has been supplimented as a #4.
R3: The author did not responsed to the reviewer's comment point-by-point.
A: The point-by-point answer file had been uploaded as a cover letter during the first response. May be the file was difficult to find.
We provide a point-by-point response to the comments as follows:
R3: Overall, this is a worthy paper containing interesting results, but the summary of this paper is relatively shallow. The views expressed are not clear enough, and the writing of the whole article is not rigorous enough. We suggest that this manuscript should be greatly revised.
A: The manuscript has been extensively revised according to your suggestion. To enhance understating of the results, the discussion section has been combined with the results. Two data has been supplemented in the Figures 3 (biofilm formation) and 6 (diameter of the hydrolyzed zone).
R3: "1.Introduction", Line 40-56 This description of gluten allergy in wheat should be introduced.
A: Thank you for your kind comment, the generation of immunogenic peptide and its symptoms has been more described on the lines 46-50.
R3: 2.4. Analysis of bacterial carbohydrate fermentation and enzyme activity. It should have the optimized design experiment.
A: The bacterial carbohydrate fermentation and enzyme activity were screened to estimate medium composition for bacterial growth and further utilization. In our study, API 50 CHB/E medium kit and API ZYM kit are commonly used in several literatures, it could checked through attached ‘doi’ address below.
https://doi.org/10.1007/s10482-019-01345-w
https://doi.org/10.1080/09593330.2018.1468492
R3: 3.5. Decomposition patterns of gluten proteins and the presence of immunogenic peptide. Please give us the reason about immunogenic peptide.
A: The section “3.6. The presence of immunogenic peptides” has been extensively revised. According to the data sheet of company, the anti-gliadin antibody, which we used in the western blot analysis, reacts specifically with the deaminated peptide (KLQPFPQPELPYPQPQ), and it comprises the 33-mer sequence part of the wheat gliadin (residues 56-88). Therefore, the reacted bands in the western blot result can be defined as immunogenic peptide. As you well known, the peptides alone can not be detected on the SDS-PAGE because of their small molecular size. However, we will check the peptides by LC with synthetic peptides, during isolation and purification of the specific enzymes in the further study.
Thank you for your consideration.
Reviewer 4 Report
Comments and Suggestions for Authors
The authors have thoroughly revised the Manuscript with a lot of changes and modifications. Therefore, I revert my decision and accept the revised Manuscript for publication in the Microorganisms Journal. Best wishes.
Author Response
Dear Reviewer 4,
We are grateful to the reviewer for the valuable comments.
Round 3
Reviewer 1 Report
Comments and Suggestions for Authors
The third version of the manuscript is essentially like the second one. The authors have not introduced new data related to the issues raised in my previous report.
Therefore my criticisms remain intact, as well as my judgment that the manuscript is preliminary, and the potential use of the activity detected in Bacillus amyloliquefaciens for diminishing allergens in food materials is speculative.
Comments on the Quality of English LanguageMinor edits are required
Reviewer 2 Report
Comments and Suggestions for Authors
The authors have done work on the errors and the article can be published in this form.
Reviewer 3 Report
Comments and Suggestions for Authors
It should improve the Language expression.
Comments on the Quality of English LanguageIt should improve the Language expression.